# phylotaR: An Automated Pipeline for Retrieving Orthologous DNA Sequences from GenBank in R

**DOI:** 10.3390/life8020020

**Published:** 2018-06-05

**Authors:** Dominic J. Bennett, Hannes Hettling, Daniele Silvestro, Alexander Zizka, Christine D. Bacon, Søren Faurby, Rutger A. Vos, Alexandre Antonelli

**Affiliations:** 1Gothenburg Global Biodiversity Centre, Box 461, SE-405 30 Gothenburg, Sweden; daniele.silvestro@bioenv.gu.se (D.S.); alexander.zizka@bioenv.gu.se (A.Z.); christinedbacon@gmail.com (C.D.B.); soren.faurby@bioenv.gu.se (S.F.); alexandre.antonelli@bioenv.gu.se (A.A.); 2Department of Biological and Environmental Sciences, University of Gothenburg, Box 461, SE-405 30 Gothenburg, Sweden; 3Naturalis Biodiversity Center, P.O. Box 9517, 2300 RA Leiden, The Netherlands; hannes.hettling@naturalis.nl (H.H.); Rutger.Vos@naturalis.nl (R.A.V.); 4Gothenburg Botanical Garden, Carl Skottsbergsgata 22A, SE-413 19 Gothenburg, Sweden; 5Department of Organismic and Evolutionary Biology, Harvard University, 26 Oxford St., Cambridge, MA 02138 USA

**Keywords:** BLAST, DNA, open source, phylogenetics, R, sequence orthology

## Abstract

The exceptional increase in molecular DNA sequence data in open repositories is mirrored by an ever-growing interest among evolutionary biologists to harvest and use those data for phylogenetic inference. Many quality issues, however, are known and the sheer amount and complexity of data available can pose considerable barriers to their usefulness. A key issue in this domain is the high frequency of sequence mislabeling encountered when searching for suitable sequences for phylogenetic analysis. These issues include, among others, the incorrect identification of sequenced species, non-standardized and ambiguous sequence annotation, and the inadvertent addition of paralogous sequences by users. Taken together, these issues likely add considerable noise, error or bias to phylogenetic inference, a risk that is likely to increase with the size of phylogenies or the molecular datasets used to generate them. Here we present a software package, phylotaR that bypasses the above issues by using instead an alignment search tool to identify orthologous sequences. Our package builds on the framework of its predecessor, PhyLoTa, by providing a modular pipeline for identifying overlapping sequence clusters using up-to-date GenBank data and providing new features, improvements and tools. We demonstrate and test our pipeline’s effectiveness by presenting trees generated from phylotaR clusters for two large taxonomic clades: Palms and primates. Given the versatility of this package, we hope that it will become a standard tool for any research aiming to use GenBank data for phylogenetic analysis.

## 1. Introduction

The first step in any nucleotide-based phylogenetic analysis is the identification of sequence homology. Without establishing homology, much like comparing apples and oranges, multiple sequence alignment is meaningless. More precisely in the context of species trees, sequences chosen for phylogenetic analysis must represent the same ancestral region resulting from a speciation event, i.e., they must be orthologous [1,2]. Sequence orthology is often determined through name-based searches via large sequence databases, most commonly GenBank [3]. This approach, however, can be problematic due to the possibility of sequences being mislabeled and differences in naming conventions. For example, gene names can differ between working groups (e.g., COI, CO1, COX1 and COXI); different sections of a gene or region may be deposited under the same sequence name [4]; and deposited sequences may represent multiple genes in what are termed chimeric sequences [5]. In the best-case scenario, these issues may lead to the failure to identify all relevant orthologous sequences. Worst case, one or more of the downloaded sequences will represent different ancestral regions, causing poor alignment and/or incorrect inference of phylogenetic trees. Without resolving the problem of orthology in a programmatic fashion, any large-scale attempt at self-updating, automated pipelines and initiatives for constructing phylogenies, e.g., [6,7], are bound to fail [8].

In an early attempt to address these issues, Sanderson et al. [4] developed a pipeline, PhyLoTa, that uses the Basic Local Alignment Search Tool (BLAST [9]) to identify orthologous sequences without the need for gene name matching. For a given taxonomic group, PhyLoTa searches through available sequences on GenBank and identifies orthologous sequence clusters. Users are then able to survey the clusters via a web-interface [10] and select the ones that best suit their phylogenetic analysis needs, e.g., by selecting the clusters that maximize coverage of their taxonomic groups of interest. A downside of PhyLoTa is that the searching and clustering is performed via all-versus-all BLASTing, the combinatorics of which become prohibitive above a certain taxonomic level—an ever increasing barrier as public sequence databases grow. One solution is to perform the BLASTing within taxonomic groups, leading to potentially shared clusters among taxonomic groups remaining undiscovered by PhyLoTa.

More importantly, however, the current PhyLoTa release is outdated as it was built on a GenBank release in February 2013 (representing 162,886,727 sequences, Release 194 [11]). Since then over 44 million new sequences have been deposited in GenBank (Release 224, representing 207,040,555 sequences, [11]). Additionally, NCBI’s taxonomic database has been updated as new sequences are added, species are discovered and groupings are revised. Between March 2013 and February 2018, 170,000 new nodes of the database’s taxonomic tree were added [12], representing 30% of all current nodes. Clearly, more frequent and regular updates to the pipeline are needed for phylogeneticists to make use of newly acquired and improved data. More recently, new databases of orthologous sequences have become available [13,14]. These databases, however, are not based on GenBank but instead on assembled genomes—massively limiting their taxonomic coverage.

To date, there have been just two alternatives for those who wish to discover orthologous sequences from GenBank: Rely on error-prone gene names, or make do with outdated information. Here we present phylotaR, an R package comprising a pipeline for identifying and retrieving orthologous sequence clusters directly from the latest GenBank release. Our pipeline recreates the original PhyLoTa output for specific clades of interest to the user in a series of independent stages. Additionally, a user has the option of a secondary cluster stage (cluster^2^) to identify and merge clusters at higher taxonomic levels than is available with PhyLoTa. We demonstrate and test the capacity of phylotaR by generating phylogenetic trees for two model clades, widely studied in evolutionary biology: palms and primates.

## 2. Implementation

### 2.1. The Pipeline

The phylotaR pipeline consists of four automated, independent stages: **taxise** (identify all descendant taxonomic nodes), **download** (hierarchically retrieve all sequences from across the taxonomic tree), **cluster** (identify clusters from the downloaded sequences within nodes) and **cluster^2^** (merge clusters identified within separate taxonomic nodes to identify clusters at higher taxonomic levels) (see Appendix A for a conceptual outline). At a minimum, all a user needs to do is provide a taxonomic identity (name or NCBI ID at any taxonomic level), for which they would like to generate sequence clusters, and then run the phylotaR pipeline. The pipeline mimics the original PhyLoTa but with the following improvements: (i) It makes use of sequence feature information to break up large sequences which may have otherwise been discarded for being too long; (ii) it can generate paraphyletic clusters from nodes which are too small in themselves and (iii) it has the additional stage for matching sister clusters, cluster^2^, which makes our method scalable to larger groups of taxa with many sequences available (see Table 1 for a comparison of phylotaR and PhyLoTa). For more details on the pipeline see Figure 1 for an outline of the process, refer to Appendix B for a detailed description of each stage, Appendix A for a description of all the parameters and Appendix A for benchmarking of how different parameters impact run-times and results.

After the phylotaR pipeline stages have completed, the user can interrogate the identified clusters using a series of functions supplied within the phylotaR package. For example, a user can filter the sequences across the clusters to a given taxonomic rank, or select sequences with clusters using a range of factors: sequence lengths, GC-ratios, sequence definitions, proportion of ambiguous nucleotides and/or maximum alignment density (MAD score, [4]). Additionally, plotting functions allow the user to see which taxa are covered by which clusters (for examples, see figures below). After exploring, modifying and/or manipulating the clusters, the user can export them in tabular format as per the PhyLoTa database schema or as sequence files in FASTA format [15], which can be readily aligned by different software.

### 2.2. Installation, Features and Future Developments

The development version of the phylotaR package is currently available via GitHub (github.com/AntonelliLab/phylotaR) and can be installed through the R package devtools [16]. It will soon be available via CRAN [17] and later Bioconda [18]. The package depends on R (v 3+) and on a range of R packages [19,20,21,22,23,24,25,26], and requires stand-alone BLAST. Instructions for installing the BLAST+ suite can be found via NCBI [27].

The entire pipeline can be run from an R console with just a few lines of code (Figure 2). The package comes with tools that enable a user to halt the pipeline process at any point. All downloaded and BLAST results are automatically cached, allowing a user to restart the pipeline after halting without loss of data. A log file records any user interventions and all pipeline progress, thus, increasing reproducibility of results and facilitating the methodological description in scientific publications. Finally, the code is developed to be modular, allowing users to contribute additional modules, functions and/or improvements. All internal functions and classes are documented to this end. To maximize the transparency and stability of the phylotaR package we have submitted the package to ROpenSci [28]—a community for ensuring good R coding practices in reproducible research—for which it is currently under review.

For future versions of phylotaR we envisage a range of additional features. For example, the ability to identify clusters across disparate taxonomic groups using the cluster^2^ stage; the incorporation of a user’s own unpublished sequences or the specification of their own taxonomy that are independent of NCBI; the incorporation of alternative search tools other than BLAST that may be faster or sequence specific (e.g., DIAMMOND [29], usearch [30], BLAT [31], PLAST [32]); a download feature that would allow a user to make use of FTP copies of large sections of GenBank to avoid slow-querying via Entrez [33]; and the ability to send BLAST queries via NCBI’s online BLAST [34] and other servers. Users can request new features and highlight bugs via the phylotaR GitHub page (github.com/AntonelliLab/phylotaR).

## 3. Empirical Demonstration: Palms and Primates

Here we demonstrate and test the phylotaR pipeline’s ability to identify orthologous sequences by constructing phylogenetic trees from clusters generated for palms (*Arecaceae*, TaxID: 4710) and primates (*Primates*, TaxID: 9443) using default parameters, see Appendix A. For palms and primates, respectively, we identified 449 and 1021 clusters, each containing over ten unique taxonomic identifiers. Taken together, these clusters included 1238 and 653 unique taxonomic identifiers and 13,344 and 56,112 sequences, respectively. (See Appendix A for visual representations of the relative distributions of sequences/taxa/clusters among the different clusters and taxa.) The runtimes for palms and primates took in total 1.03 and 4.11 h, respectively. The download stage took by far the longest to complete, taking 0.89 and 3.47 h for palms and primates, respectively, representing 86% and 84% of total runtime.

Because phylotaR makes use of a more recent GenBank release than PhyLoTa, more sequences, taxa and clusters can be identified by phylotaR than PhyLoTa. We can compare the phylotaR results presented here to the equivalent results generated by PhyLoTa. According to the PhyLoTa browser [10] for palms there are 98 ‘phylogenetically informative clusters’—i.e., clusters with more than four unique taxonomic identifiers. In the phylotaR results, there are 1011 phylogenetically informative clusters. Comparing the clusters with the most taxonomic identifiers between the two methodologies, the PhyLoTa browser identifies 648 taxonomic identifiers, 926 sequences and 164 genera. The top cluster in the phylotaR results for palms identifies 720 taxonomic identifiers, 1346 sequences and 160 genera. The differences are even greater for the primates. 3727 clusters with more than four unique taxa were identified by phylotaR, whereas the equivalent figure for PhyLoTa is 1103. Comparing the clusters with the most taxonomic identifiers, PhyLoTa has 129 taxa, 543 sequences and 49 genera while phylotaR has respectively 491, 4871 and 75. (See, respectively, Appendix A for comparisons of clusters identified for palms and primates by PhyLoTa [S3a and S4a] and phylotaR [S3b and S4b].).

Despite these differences in numbers, many of the same sequences identified by PhyLoTa were identified by phylotaR. For example, we were able to identify equivalent clusters between PhyLoTa and phylotaR by matching sequence identifiers. We found that across the clusters for palms identified by PhyLoTa, a mean 92.02% of the sequences in any of the clusters could be matched to sequences in one of the clusters identified by phylotaR. This figure was much smaller for primates. After controlling for differences in taxonomy, sequence length, and model organisms (for which sequences are selected randomly), only 55.12% of PhyLoTa sequences could be matched to sequences in phylotaR clusters. These ‘missing’ sequences in the phylotaR clusters were found to have either new sequence identifiers (10%), to be clustered in clusters with fewer than four sequences (30%) or to have not been in any clusters (60%).

To illustrate the ability of the pipeline to correctly identify orthologous sequence clusters while keeping the demonstration fast, we generated small, representative trees for both palms and primates at the tribe and genus levels, respectively. We reduced the number of sequences within each cluster to just representatives within these taxonomic levels by filtering the sequences by proportion of ambiguous nucleotides and length of sequence. We then selected the ‘best’ clusters for both palms and primates from these reduced clusters, by dropping all clusters with sequence maximum alignment densities (MAD, [4]) less than 0.75 and then selecting the top ten with the greatest number of tribes/genera (see Appendix A for detailed information on each of the selected clusters). These top clusters were found to be representative of palms and primates as a whole (Figure 3).

For each of the clusters, the sequences were written out in FASTA format and alignments were constructed using MAFFT (v7.271, [35]) with its *--auto* argument. Supermatrices were then constructed using the ten sets of alignments. RAxML (v8.2.4, [36]) was then used to construct the trees. We used the GTRGAMMA model and partitioned the supermatrices based on the identified clusters. We ran a rapid bootstrap analysis of 100 iterations. To monitor for errors at any point in the demonstration pipeline two sequences were selected instead of just one when reducing the clusters down to tribe/genus level. If in the resulting trees, large numbers of representative sequences of the same taxon were not uniquely clustered, data and methods were re-checked and the pipeline was run again. For palms and primates, sequences in the same tribe/genus were sisters in, respectively, 100% and 94% of cases.

We found the generated trees to be similar to those published for these groups (Figure 4, Appendix A) with few differences in topology (for a detailed assessment of the trees, see Appendix C). Although it is likely that our post-phylotaR pipeline for constructing the trees could be improved to account for these topological differences, by successfully recreating the overall phylogenetic relationships of these groups we have demonstrated that phylotaR is able to successfully identify orthologous sequences. Additionally, we would expect no major topological differences to equivalent results generated from PhyLoTa data as, respectively for palms and primates, a mean 99.1% and 99.9% of sequences across the best-matching PhyLoTa clusters were found in the phylotaR clusters used for generating the trees.

The methods described above and in the appendix can be reproduced via scripts available from GitHub (github.com/AntonelliLab/phylotaR_demo).

## 4. Conclusions

The phylotaR package offers a user-friendly pipeline to obtain orthologous sequences for phylogenetic inferences from GenBank in R. The phylotaR pipeline extends upon the PhyLoTa framework by, in addition, making use of sequence feature information, generating small paraphyletic clusters to prevent dropping taxa and implementing a subsequent cluster^2^ to identify shared clusters across large taxonomic groups. The phylotaR pipeline, as shown here, can be used to yield reliable results for phylogenies of large taxonomic groups. The pipeline is modular and can be easily integrated into R workflows. We envisage phylotaR to be an important first step and module as part of an ecosystem of current and future automated, phylogeny-generating platforms [8]. The package is currently available via GitHub (github.com/AntonelliLab/phylotaR) and comes with detailed vignettes containing documentation and tutorials.

## Figures and Tables

**Figure 1 life-08-00020-f001:**
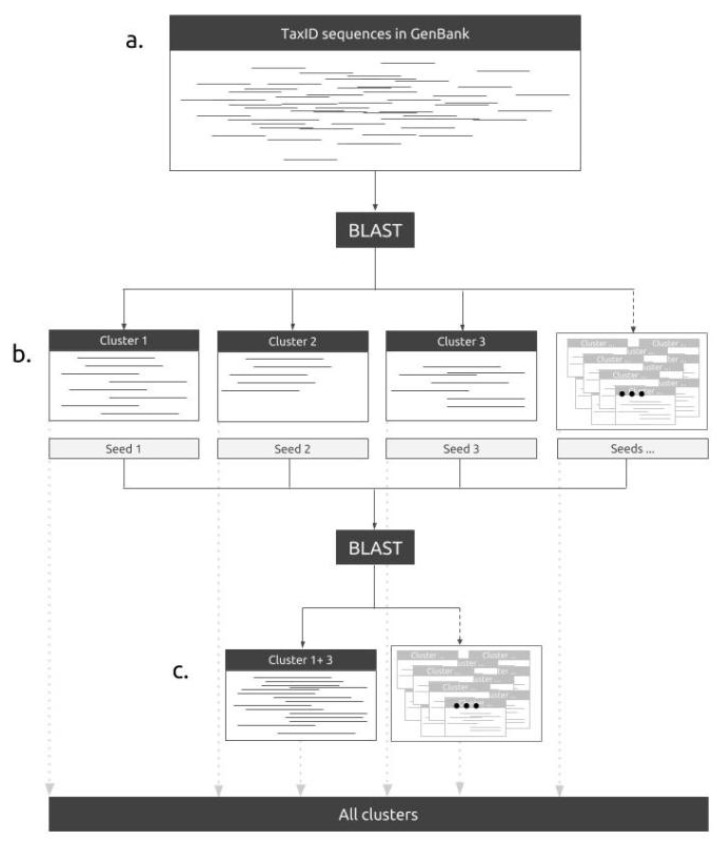
The phylotaR pipeline identifies all sequences in GenBank associated with a user-specified taxonomic identity (**a**). The pipeline then performs all-vs.-all BLAST across all the sequences to identify orthologous clusters (**b**). These searches are constrained to run within taxonomic groups up to a user-determined limit (default 50,000 sequences and 100,000 nodes). To generate higher taxonomic level clusters, an additional BLAST search is performed of the most connected sequences within clusters (i.e., the seed sequences) from the lower-level clusters. The clusters of overlapping seed sequences are then merged into larger clusters (**c**). All clusters, merged and non-merged, are then reported for inspection by the user. For more details on the pipeline, see Appendix B.

**Figure 2 life-08-00020-f002:**
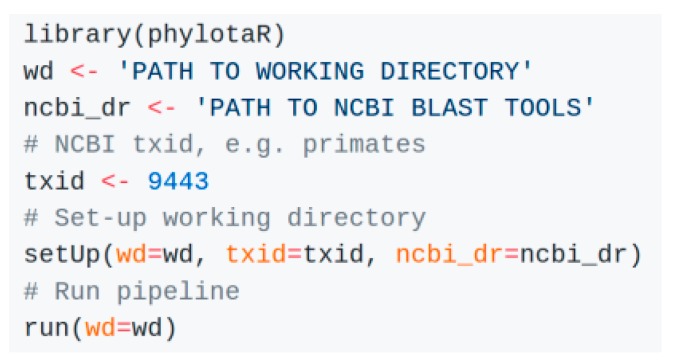
Initiating the phylotaR pipeline in R for primates (TaxID: 9443).

**Figure 3 life-08-00020-f003:**
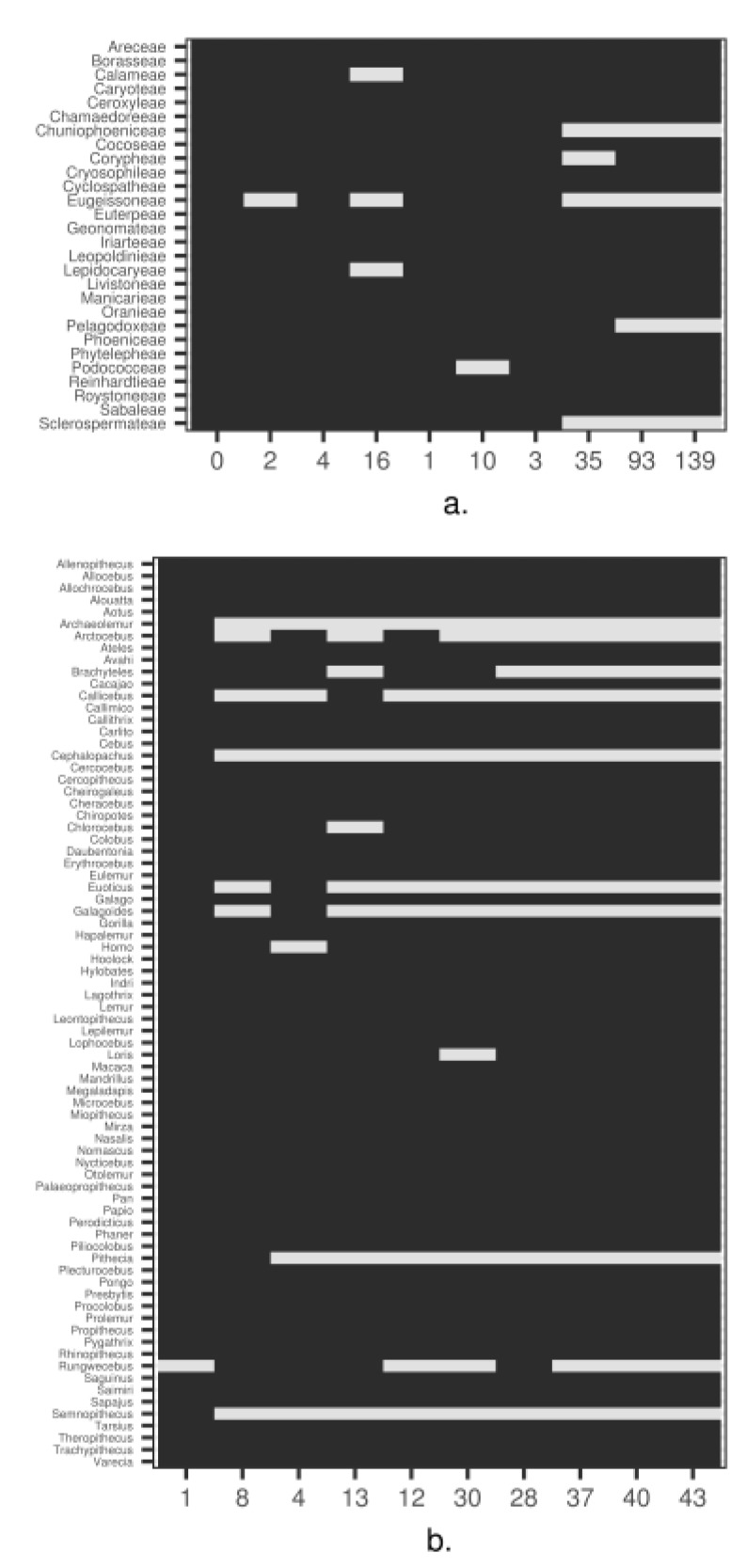
Presence/absence of tribes and genera for palms (**a**) and primates (**b**), respectively, across the top ten best clusters. X-axis numbers are unique cluster Ids. For more details on each of these clusters, see Appendix A.

**Figure 4 life-08-00020-f004:**
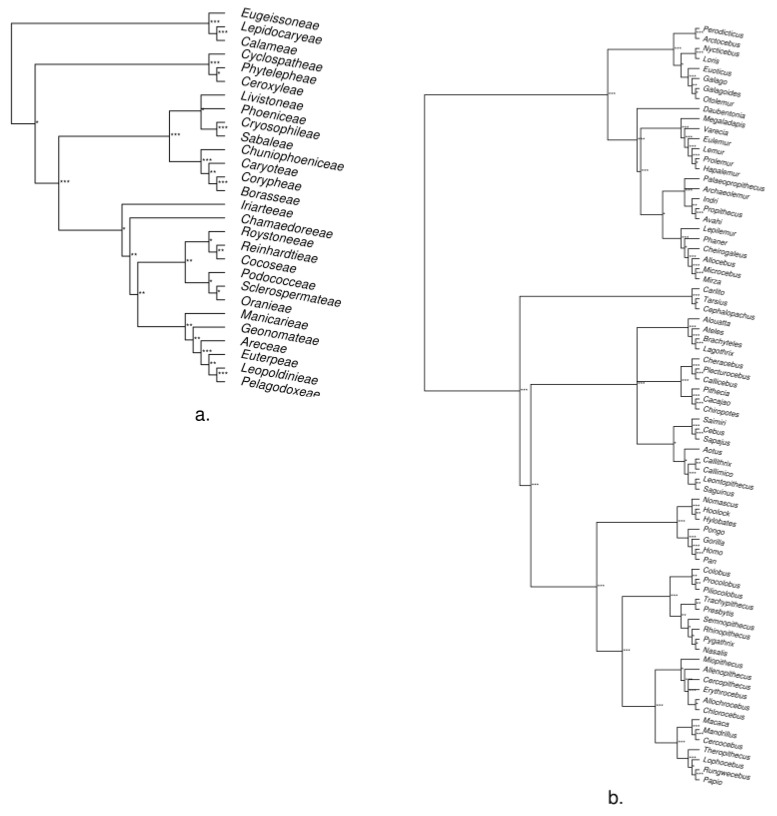
Tribe- and genus-level trees for palms (**a**) and primates (**b**). Roots were determined manually by rooting with *Strepsirrhini* and *Calamoideae* for primates and palms respectively. Branch lengths have been removed. Support calculated from 100 rapid bootstraps: *** >0.95, ** >0.75 and * >0.50. Complete tree construction methods are in Appendix C. For tree comparisons with published trees for palms [37] and primates [38], see Appendix A.

**Table 1 life-08-00020-t001:** Comparing phylotaR and PhyLoTa.

	phylotaR	PhyLoTa
*Features*		
Direct clades	Yes	Yes
Subtree clades	Yes	Yes
Paraphyletic clades	Yes	No
Merged clades	Yes	No
Outputs	Clusters	Clusters, alignments, trees
*Implementation*		
Language	R	Perl
Open source	Yes	No
Execution	Local computer	Web-interface
Modular design	Yes	No
*Resources*		
GenBank release	Latest	2013
Search-tool	BLAST, user-choice *	BLAST
Taxonomy	NCBI, user-choice *	NCBI
Sequence features	Yes	No
Non-NCBI sequences	Yes *	No

* Yet to be implemented features.

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
