# Peer review of "phylotaR: An Automated Pipeline for Retrieving Orthologous DNA Sequences from GenBank in R"

_life, 2018, doi:10.3390/life8020020_

Round 1

Reviewer 1 Report

  Authors proposed a systematic method to retrieve orthologous DNA sequences from GenBank in R. They also demo their methods by some examples. 

  Yet, their contribution only focus on the data cleaning part for phylogenetic analysis. 

  It would be better they could try to apply their methods to more data set. 

Author Response

We thank the reviewer for taking the time to review our paper.

Only the beginning of the phylogenetic pipeline

We understand that our pipeline is only focussed on the beginning of a phylogenetic analysis. This was the intended aim of our R package and we have no intention to increase its scope any further. In the past, many of us have developed programs that do span a complete phylogenetic analysis. Such programs are, however, difficult to maintain, rigid and go out of date very quickly. As a new approach, we are attempting to develop all the key modules of a complete phylogenetic pipeline as separate, maintainable packages. We hope phylotaR to be an important early module in our own future phylogenetic pipelines and work-flows, as well as those of others.

More datasets

We argue that our empirical examples are sufficient in demonstrating phylotaR. The objective ofthephylotaR pipeline is to identify orthologous sequences. In being able to create trees that are similar to those already published for large, well-cited and studied clades indicates that our pipeline process is able to achieve this objective. Additionally, the main functionality of our pipeline is not sufficiently different from that published by Sanderson et al. (2008) as part of PhyLoTa. To better elucidate our point, we have clarified and expanded parts of the main text and have added new tables to the supplementary materials for comparing phylotaR and PhyLoTa output for our two demonstration taxonomic groups.

p { margin-bottom: 0.25cm; line-height: 120%; }a.cjk:link { }a.ctl:link { }

Reviewer 2 Report

The paper presents a pipeline, phylotaR to identify orthologous sequences. The pipeline builds on an existing tool, the PhyLoTa browser, updating it to more recent version of GenBank and extending it with new features. The code is released as open source on github and reviewed for quality. Additionally, the package is under consideration for CRAN, which will facilitate its adoption even further.

The paper is well written, in clear and correct language and style. Figures and tables are well used to clarify the method and assessment of results. The paper is a pleasure to read, however I missed some methodological elements that are normally presented when one proposes a new pipeline:

1)  The authors honestly say that they ‘build on PhyLoTa’. What does this mean exactly? Has PhyLoTa served as inspiration, did you reuse the methods and rewrite the code, or could some of the code be reused? Exactly which parts were improved and which remained the same?

2)  How does phylotaR compare to PhyLoTa? In particular, are the results different, comparable or somehow better? An explicit comparison should be included and discussed in the paper (as much as possible, due to intrinsic differences)

3)  I would have expected more emphasis in the assessment and validation of the new pipeline, instead of just an ‘empirical demonstration’. For the potential new users of phylotaR it would be important to know why they should trust this code – and when. I therefore would suggest to move part of appendix B into the paper body, and also expand the discussion. In particular the results of primates seem to be of lower quality than those for palms – is this somehow to be explained by some intrinsic property of the method or the test case? How generalizable are the conclusions about the quality for other cases?

Finally, could you also add some information about the computational effort and requirements for running the pipeline? Is there any bottleneck that could affect the quality of results? For example, is any parallelism exploited, e.g. during bootstrapping?

Typos:

• programtic
• BLASTingwithin

Author Response

Clarifying ‘built upon PhyLoTa’

The programmatic process that the phylotaR pipeline performs is based on PhyLoTa as described in the original manuscript (Sanderson et al. 2008). phylotaR builds on this core pipeline with additional features. The programs are entirely distinct and are even written in different programming languages. In developing phylotaR, we have not even looked at any of the original PhyLoTa code. To better clarify the differences between phylotaR and PhyLoTa, we have adapted the text in the introduction and the implementation and we have added Table 1 to the manuscript.

Comparing output from PhyLoTa and phylotaR

To give the reader confidence in our method and to better compare PhyLoTa and phylotaR, we have added a new table in the appendix comparing the results from phylotaR and PhyLoTa for palms and primates.

Phylogeny accuracy and testing the phylotaR pipeline

The purposes of the empirical demonstrations were to simply show that good trees could be constructed from phylotaR-determined clusters, indicating that phylotaR does indeed identify orthologous sequences. In this sense, by constructing trees from phylotaR output, we are in effect testing the pipeline. We do not make any claims on the phylogenetic relationships of the palms and primates. So long as any oddities and errors in the phylogenies are not the result of phylotaR itself, we would argue that these are not of supreme importance for our aims and goals in this manuscript. Given that the phylogenies are as close as they are to published phylogenies, it would seem that phylotaR must have accurately identified orthologous sequences. Our phylogenies could potentially be improved by developing upon our post-phylotaR process that involves: selection of phylotaR clusters, the alignment process and phylogenetic inference.  We feel that doing so, would detract from our efforts to develop phylotaR and would, potentially, open new debates on the specifics of palm and primate phylogenetics.

As per the reviewer’s comments, we have moved large parts of appendix b to the methods and have expanded the discussion in the main text and the appendix.

Runtime

We apologise for omitting to make mention of the demonstration runtimes. Upon looking up the log files, we find that the palms pipeline began at 14:08:21 on 2018-03-22 and, taking 1.03 hours, finished without pauses at 15:10:09. The primates pipeline began at 14:08:12 on 2018-03-22, it had one server error at 15:37:48 and was restarted at 16:50:05. It then continued without issue and completed on 19:27:15, taking in total 4.11 hours. The longer runtime for the primates is due to the larger amount of available sequences. We have added these times to our manuscript.

Bottlenecks

The biggest runtime bottleneck is the download phase. The palms and primates download steps took 0.89 and 3.47 hours, respectively, representing 86% and 84% of total runtime. To reduce runtime, we are currently developing an R package that downloads whole sections of GenBank from which a local database can be created and queried (https://antonellilab.github.io/restez/).

No parallelism is currently used – although there does exist an argument for it – but we will likely make use of the parallelism built within the BLAST program (and/or alternative search tools when they are implemented).

No bootstrapping is performed by phylotaR. Bootstrapping was performed with RAxMl as part of the empirical demonstrations.

Typos

These have been corrected, thank you.

Reviewer 3 Report

Dear authors,

I read your paper and would like to point out some parts that might require your attention.

First of all, I think the method itself is scientifically sound and provides functionality of interest to the general audience the tool is intended for. As such, I believe the method can be indeed seen as easy to apply and does not require more than basic R knowledge to be run and installed on a system.

Therefore, I only have some points that could improve the general paper / method for users by improving certain aspects of the pipeline.

First of all, I noticed that you utilize BLAST(+) to perform the cluster process, which I believe can be improved significantly. There are tools such as DIAMOND https://github.com/bbuchfink/diamond  that are able to perform the same kind of searches and perform this at ~500-2000x the speed of BLAST. You should at least consider adding this as an alternative algorithm instead of BLAST and/or have this as a planned feature for an update of your pipeline.

Aiming in the same direction, this would also enable you to perform an evaluation of BLAST and DIAMOND within such a phylogenetic context, ultimately providing the audience with a more sophisticated information on whether one or the other of the two provided algorithms in your pipeline is suited better for their particular type of analysis. This is just a suggestion, but I'd see this as a significant improvement of the entire pipeline and the paper!

My second point would be to provide a (bio-) conda package for your entire pipeline and its dependencies (https://bioconda.github.io/ or https://github.com/bioconda/bioconda-recipes/ ).

This would enable users to utilize your pipeline on complex cluster/HPC environments as well as on cloud providers or local workstations without the requirement to install the methods in a complicated way. I'm not against a local installation, more just suggesting that this could indeed increase both the visibility of your tool (as part of bioconda packages) and therefore increase your audience, which I believe is in your intent anyways! Extra Bonus when you submit a bioconda package: You will automatically get Docker and Singularity containers of your pipeline for users to download, that will further be of use to users having issues to install software e.g. on their computing environments.

Can you also elucidate a bit more where you see similarities of the SUPERSMART pipeline you mentioned in the paper and evaluate this a bit more? The current paper does not show this explicitly and I think it would benefit quite significantly from an added evaluation of SUPERSMART vs phylotaR (at least as far as these can be compared).

Typos:

l164/165: (in Figure 4: The genus names should be in italics).

Author Response

We thank the reviewer for providing such a positive and actionable review. We very much appreciate the time and thought they have put in to reviewing our manuscript and hope that our responses meet their expectations.

BLAST

At this stage, we decided not to implement any alternatives to the BLAST search algorithm. Although often slower, the big advantage BLAST has over alternatives is it is a general nucleotide search tool and can work with all sequence types. We looked into using DIAMOND, but discovered that it currently only works with protein or reverse translated sequences. (But it looks as if this might change: https://github.com/bbuchfink/diamond/issues/117). We also considered other search tools (e.g. usearch, BLAT, PLAST ….) but decided that these also were more specific than BLAST, either in terms of sequence type or in computational environment. Additionally, many of these tools, while faster, may be less sensitive. Finally, we would like to point out that in both of our demonstrations and in other phylotaR runs that we have performed, by far the biggest limiting factor was not the BLASTing but rather the downloading. In order to speed up phylotaR, we have already begun work on a new R package for downloading whole sections of GenBank (https://antonellilab.github.io/restez/) that should massively speed up the pipeline runtime.

All of the above being said, we have highlighted alternatives to BLAST as a potential future feature on our github page (https://github.com/AntonelliLab/phylotaR/issues/33). Additionally, we have updated the text within our manuscript to make mention of the idea that alternative search-tools could be used. Currently, we are considering a general, search-tool approach where a user can develop their own input and output functions for the clustering stages. The advantages of implementing alternatives to BLAST in this fashion would be: reduced need to develop code for every version and OS for each search tool, no need to ensure the phylotaR code can implement the latest search tool. By default, phylotaR would always use BLAST.

Bioconda

We really like the idea of providing our R package as a bioconda package and are grateful for the recommendation. We would like to first, however, meet the requirements for ROpenSci and CRAN. Upon inspecting the procedure for uploading a package to bioconda, it seems that is possible to directly transfer a CRAN package to bioconda (https://bioconda.github.io/guidelines.html#r-cran). This would be our preferred process: ensure R code is up to R code repository standards, then transfer to non-R code repositories. We would for example, copy the recipe set-up as implemented by gsmoothr (https://bioconda.github.io/recipes/r-gsmoothr/README.html). We have added bioconda to our mani manuscript text.

SUPERSMART

SUPERSMART is a general, programmatic pipeline for constructing large phylogenetic trees. We are the group that developed SUPERSMART, but feel that we need to take its development into a different direction. We have decided to make the process far more modular and publish each of its steps as separate R packages. PhylotaR is the first of these R package modules. In this paper, however, we do not intend to ‘announce’ our future vision for an updated SUPERSMART, only to make reference to it as a pipeline that makes use of PhyLoTa. To avoid confusion, we have removed the explicit mention of SUPERSMART.

Typos

Genus names now in italics for figure 4.

Round 2

Reviewer 2 Report

The authors have improved the manuscript and addressed some of my previous remarks. Nevertheless, the presentation of performance of the new method still could be improved.

First (and repeating myself), it would be useful to present results of more thorough performance assessment, instead of a demonstration. For example, what is the impact of parameter choices for the method both in result quality and in execution time? Are results robust under variations, e.g. computer platform or OS? Such an assessment would contribute to build trust on the tool and enable it to be picked up more rapidly by the community. 

Second, a deeper discussion about the differences between results of the old and the new method should be included. Now the authors do not make an effort to help the reader assess the differences, as the results are presented separately. For example, the authors could apply the same 'post-processing' pipeline to both results and compare the obtained trees, which seem more interpretable than the loose tables. Moreover, the authors say:

"Although identifying more clusters and from many more sequences, the clusters that phylotaR identified and comparisons of ‘phylogenetically informative clusters’ identified for palms and primates by PhyLoTa [S2a and S3a] and phylotaR [S2b and S3b].)"

So what is the gain of the new method? And where does the gain (possibly) come from? Could you show the advantage of using a newer GenBank release, for example with some special case?

Detailed/minor suggestions:

+ the captions in the supplemental materials need to be more descriptive. Also, comparative information should be compiled into one figure or table.

+ Table 1:

Architecture -> Implementation

Implementation -> Execution

typos:

+ bioconda -> Bioconda

+ Although, it is likely -> Although it is likely 

+ ortholgous sequences

+  work-flows ->  workflows

+ future current automated -> future automated

Author Response

Thank you for taking the time again to review our manuscript. We have made changes as per your recommendations.

Performance assessment
We have benchmarked the pipeline for the palms by running the pipeline multiple times under different parameter choices. The differences in run-times across the different stages and statistics on the results are recorded in a new supplementary table (Table S2). We did not run the analysis on the primates because of the much longer run-time and the greater chance of the timings being impacted by factors outside of our control (e.g. internet connection, server errors).
To briefly sum up the results of the benchmarking, the majority of the parameters have little impact on the overall results, the most important being the maximum and minimum sequence lengths. By default we select lengths above 250 and below 2000 nucleotides. As demonstrated in the new table, expanding these limits leads to be more sequences being downloaded and more clusters identified. We feel that sequences below 250 are likely not very phylogenetically informative, while too large sequences, above 2000, are more difficult to align. Ultimately, however, these choices are up to the user and will depend on their analysis.
For clarification, the pipeline works equally well across OSs and has been tested on UNIX and Windows.

Comparing PhyLoTa and phylotaR
We have generated a more detailed analysis comparing results for palms and primates from phylotaR and PhyLoTa which we describe in more detail in our manuscript. The biggest advantage of phylotaR over PhyLoTa is the newer GenBank release has more data, this fact is much better reflected in our description comparing the two methodologies. Additionally, we point out that it is unlikely that the resulting trees would differ significantly topologically because the data that PhyLoTa has, phylotaR has too.

Minor suggestions
Supplemental – We have updated the captions, but have not merged ‘comparative information’, which we think refers to Tables S2 and S3. We think these should remain separate as they are the way information is portrayed in the PhyLoTa web browser. Furthermore, the main conclusions from these tables (number of clusters, sequences etc.) are not stated in the manuscript.
Table 1 – Updated
Typos – Thanks for pointing these out, they are now corrected.

Reviewer 3 Report

Dear authors,

I feel like you adressed most points that were raised and are at least planning to add some of the requested ideas/features in an upcoming version of the method, so I feel happy about this being published soon.

Thanks for your effort!

Author Response

Thanks very much!

Round 3

Reviewer 2 Report

Thanks to the authors for addressing my comments constructively